# PPARγ Modulators in Lung Cancer: Molecular Mechanisms, Clinical Prospects, and Challenges

**DOI:** 10.3390/biom14020190

**Published:** 2024-02-04

**Authors:** Jiyun Zhang, Miru Tang, Jinsai Shang

**Affiliations:** 1School of Basic Medical Sciences, Guangzhou Laboratory, Guangzhou Medical University, Guangzhou 511436, China; zhang_jiyun@gzlab.ac.cn; 2Guangzhou National Laboratory, Guangzhou 510005, China; 3School of Pharmaceutical Sciences, Sun Yat-sen University, Guangzhou 510006, China

**Keywords:** peroxisome proliferator-activated receptor gamma, lung cancer, agonists, transcriptional activity

## Abstract

Lung cancer is one of the most lethal malignancies worldwide. Peroxisome proliferator-activated receptor gamma (PPARγ, NR1C3) is a ligand-activated transcriptional factor that governs the expression of genes involved in glucolipid metabolism, energy homeostasis, cell differentiation, and inflammation. Multiple studies have demonstrated that PPARγ activation exerts anti-tumor effects in lung cancer through regulation of lipid metabolism, induction of apoptosis, and cell cycle arrest, as well as inhibition of invasion and migration. Interestingly, PPARγ activation may have pro-tumor effects on cells of the tumor microenvironment, especially myeloid cells. Recent clinical data has substantiated the potential of PPARγ agonists as therapeutic agents for lung cancer. Additionally, PPARγ agonists also show synergistic effects with traditional chemotherapy and radiotherapy. However, the clinical application of PPARγ agonists remains limited due to the presence of adverse side effects. Thus, further research and clinical trials are necessary to comprehensively explore the actions of PPARγ in both tumor and stromal cells and to evaluate the in vivo toxicity. This review aims to consolidate the molecular mechanism of PPARγ modulators and to discuss their clinical prospects and challenges in tackling lung cancer.

## 1. Introduction

Lung cancer is the second most commonly diagnosed cancer and the leading cause of cancer-related death, with an estimated 2.2 million new incidences and 1.8 million mortalities worldwide in 2020 [1]. It can be broadly categorized into non-small cell lung cancer (NSCLC) and small cell lung cancer (SCLC), comprising ~85% and ~15% of all cases, respectively. The etiological factors of lung cancer are diverse, of which tobacco exposure is the primary risk factor, while environmental exposures such as biomass fuels, industrial carcinogens, and air pollution also strongly contribute to the development of lung cancer [2]. Unfortunately, patients with lung cancer often lack obvious specific symptoms initially and are diagnosed at an advanced stage, which might drastically reduce the 5-year survival rate from 90% (stage IA) to 10% (stage IV) [3]. Currently, the therapeutic options for lung cancer mainly include surgery, chemotherapy, radiotherapy, targeted therapy, and immunotherapy. Although multiple treatment options are available, therapeutic resistance remains a major obstacle for patients to gain continuous clinical benefits [4]. Therefore, there is an urgent need to explore more therapeutic strategies to improve the clinical outcomes of patients with lung cancer.

Peroxisome proliferator-activated receptor gamma (PPARγ), namely nuclear receptor superfamily 1 group C member 3 (NR1C3), serves as a ligand-activated transcription factor that controls the expression of genes related to lipid and glucose metabolism, energy homeostasis, cell differentiation, and inflammation [5]. Dysregulation of PPARγ target gene profiles is closely linked to tumorigenesis, as underscored by the loss of CD36 resulting from repression of the PPARγ transcriptional program in breast cancer progression [6,7,8]. Furthermore, a growing body of evidence indicates that PPARγ activation prevents cancer in tissues such as the colon, breasts, and lungs [9]. PPARγ agonists have been demonstrated to exert anti-lung cancer effects by promoting cell differentiation, inhibiting cell proliferation, and inducing cell death [10,11]. Thus, PPARγ holds great potential as a therapeutic target for tackling lung cancer. In this review, we summarize the molecular mechanism underlying the action of PPARγ agonists and highlight the role of PPARγ activation in the complex regulatory network of lung cancer, aiming to provide a reference for developing novel therapeutic strategies for lung cancer.

## 2. Structure of PPARγ

The *PPARG* gene gives rise to four transcripts by differential promoter usage and alternative splicing, which results in the production of PPARγ1 (encoded by *PPARG1*, *PPARG3*, *PPARG4* mRNAs) and PPARγ2 (encoded by *PPARG2* mRNA) isoforms [12,13,14,15]. PPARγ2 has the same sequence as PPARγ1, except for an additional 28 amino acids at its N-terminus [12]. PPARγ1 is ubiquitously abundant in many tissues, whereas PPARγ2 is preferably expressed in adipocytes.

PPARγ shares a typical NR domain structure composed of five domains: A/B, C, D, E, and F domain (Figure 1A) [16]. The N-terminal A/B domain is highly variable among the NR family, and harbors a ligand-independent transcriptional activation function region termed AF-1 that regulates PPARγ activation through interdomain coordination and phosphorylation [17,18]. The C domain, also known as the DNA-binding domain (DBD), is the most conserved part of NRs and consists of two zinc finger motifs with nine cysteines [17]. This domain specifically recognizes and binds to the PPAR response elements (PPRE) on the target gene promoter to initiate transcription after forming a heterodimer with the retinoic X receptor α (RXRα) [19,20]. The poorly conserved D domain serves as a flexible hinge that allows rotation between the DNA-binding and ligand-binding domains, as well as containing a nuclear localization signal [17]. The E domain, also named the ligand binding domain (LBD), is the largest region of PPARγ and has four main functions, including a second dimerization interface, the ligand binding pocket, a coregulator binding surface, and ligand-dependent activation function referred to as AF-2 [21]. The C-terminal F domain is relatively small, and may contribute to interaction with cofactors [22].

Among PPARγ domain structures, the LBD was the first one to be characterized in conformation [23]. The PPARγ LBD consists of 13 α-helices (termed helix 1–12 and helix 2′) arranged in a three-layered sandwich and a small four-stranded β-sheet. The ligand binding pocket is located in the center of the LBD, and has a large Y- or T-shaped cavity with three branches [21]. Branch I is composed of helix3, helix5, helix11, and helix12 in the C-terminal AF-2 region; branch II is positioned around helix2′, helix3, helix6, helix7, and the β-sheet region; while branch III is surrounded by the β-sheet, helix2, helix3, and helix5. Adjacent to the ligand-binding pocket, there is an AF-2 coregulator interaction surface formed by the three-dimensional association of helix3, helix4, helix5, and helix12 [23,24].

## 3. Transcriptional Activities of PPARγ

PPARγ regulates gene expression through transactivation and transrepression. In the absence of ligands, PPARγ-RXRα heterodimer binds to PPREs and subsequently recruits corepressors and associated chromatin-modifying enzymes to silence target gene transcription, a process known as ligand-independent repression [25]. Once the ligand binds to PPARγ, the PPARγ-RXRα heterodimer undergoes a conformational change that releases corepressors in exchange for coactivators, resulting in the transcription of target genes. Furthermore, PPARγ can also negatively regulate gene expression in a ligand-dependent manner by antagonizing other transcription factors, such as nuclear factor-κB and activator protein-1 [26].

## 4. Ligands of PPARγ

The majority of the molecular functions of PPARγ are regulated by its ligand molecules that can be grouped as natural and synthetic ligands (Table 1). The natural ligands, also termed endogenous agonists, can be further divided into four subgroups, namely the eicosanoid prostaglandin-A1 and the cyclopentenone prostaglandin 15-deoxy- D12,14-prostaglandin J2, the unsaturated fatty acids, the nitroalkanes, and the oxidized phospholipids [27]. Notably, the natural ligands are not always efficient for PPARγ activation and target gene transcription [28,29]. Synthetic ligands can be classified as full agonists, partial agonists, antagonists, and inverse agonists. A well-known example of a full agonist is thiazolidinediones (TZDs), which are recognized for their potent insulin-sensitizing effects in type II diabetes mellitus [30]. However, TZDs also cause undesired effects such as weight gain, edema, and heart failure, which has driven the development of safer PPARγ partial agonists to avoid the toxicity induced by full agonists [31]. Partial agonists, referred to as SPPARMs, retain high affinity to the PPARγ, but show reduced transcription of given genes [21]. In contrast, antagonists such as GW9662 suppress the transcription of PPARγ-responsive genes, which is achieved by competitively binding the LBD pocket with agonists [32]. T0070907, an inverse agonist with a similar chemical structure to GW9662, but a different effect on PPARγ transcriptional regulation, inhibits the transactivation potential of PPARγ below basal cellular levels by recruiting corepressors [33,34]. Generally, synthetic ligands are thought to regulate the transcription activation of PPARγ by completely displacing natural/endogenous ligands upon binding to the ligand-binding pocket in a competitive or “one-for-one” manner. Our recent research put forward a cooperative cobinding concept of endogenous and synthetic ligands to synergistically activate PPARγ, extending the understanding of nuclear receptor ligand exchange models [35].

## 5. Dynamic Mechanisms of Ligand Binding Agonism

In the absence of ligands, helix 12 acts as a highly flexible switching element in equilibrium between many different conformations, ranging from active to repressive (Figure 1B) [34,42]. Agonists stabilize helix 12 in a conformation that exposes the AF-2 surface for the binding of coregulators that control target genes transcription [43]. Full PPARγ agonists stabilize an active AF-2 surface conformation via forming a critical hydrogen bond with Y473 residues on helix 12, facilitating the recruitment and binding of coactivators [44]. Partial agonists generally do not form hydrogen bonds with the key residues of the AF-2 regions, including Y473, but mildly stabilize helix 12 through interactions with other regions of the ligand-binding pocket, resulting in differential coactivator recruitment profiles and weak transcriptional activation compared to full agonists [21,45,46]. In contrast, inverse agonists exert transcriptional repression via stabilizing in a conformation that favors the recruitment of corepressors. However, the structural mechanism underlying the ligand-induced repression state is limited. Brust et al. identified R288 as the critical corepressor-selective inverse agonist (T0070907) switch residue, and found that T0070907-bound PPARγ exchanges between two long-lived conformations, one similar to the coactivator-bound state and the other similar to the corepressor-bound state [33]. Subsequently, our study further verified the structurally diverse mechanism of the inverse agonist-bound state and revealed the mechanism of action of T0070907 [34]. Briefly, T0070907 can stabilize helix 12 within the orthosteric pocket by pointing the pyridyl group toward the AF-2 surface, thereby increasing corepressor binding affinity.

The preceding section has provided a description of the binding modes demonstrated by various agonists. Subsequently, the dynamic changes in agonist binding with PPARγ LBD will be outlined below. Prior studies remain controversial as to whether ligand binding proceeds through induced fitting or conformational selection mechanisms. Notably, our recent study supported the existence of the induced fit mechanism involving a two-step process of an initial ligand encounter complex followed by a conformational change (Figure 1B) [47]. In the absence of ligand, helix 12 in apo-PPARγ LBD exchanges between transcriptionally repressive and a solvent-exposed active conformation through entering and exiting the orthosteric ligand-binding pocket. Agonist binds to the ligand entry site via an initial fast step to form an encounter complex, and this process can occur to either of these conformations. Subsequently, the agonist slowly enters the orthosteric ligand-binding pocket and forms the final ligand-binding pose. In this step, agonist binding to the repressive LBD conformation (helix 12 within the orthosteric pocket) would push helix 12 into an active conformation, while agonist binding to the active LBD conformation (helix 12 outside the orthosteric pocket) would facilitate transition into the final ligand-binding pose.

## 6. Role of PPARγ Activation in Lung Cancer

### 6.1. Regulation of Lipid Metabolism

Metabolic reprogramming is a crucial hallmark of malignancy, allowing tumor cells to meet demands for growth, proliferation, and metastasis, as well as be robust against unfavorable environments [48,49]. Thus, targeting abnormal tumor metabolic activities, including lipid metabolism, is a rapidly emerging direction for anti-cancer therapy [50]. PPARγ is a central regulator of lipid metabolism. Several studies have shown that PPARγ upregulates fatty acid synthesis and β-oxidation in lung cancer (Figure 2). For example, Phan et al. found that pioglitazone-mediated PPARγ activation induced de novo fatty acid synthesis and β-oxidation in lung cancer [51]. Importantly, dramatic lipid synthesis could deplete nicotinamide adenine dinucleotide phosphate (NADPH), a major reducing agent important for cellular anti-oxidation systems, leading to disrupted redox balance which, in turn, suppresses lung cancer. Moreover, Andela et al. reported a shift in cellular energy metabolism towards fatty acid oxidation in the lung alveolar carcinoma cell line via treatment with PPARγ agonist troglitazone [52].

Aldehyde dehydrogenases (ALDHs) act as an ‘aldehyde scavenger’ during lipid peroxidation and exhibit high activity in lung cancer [53,54]. Inhibition of ALDHs can expose cancer cells to highly reactive and toxic aldehydes, resulting in cell damage and apoptosis [55]. Notably, PPARγ has been reported to downregulate certain members of the ALDH family to function as a lung cancer inhibitor. For instance, arachidonic acid-induced PPARγ activation suppressed the growth of A549 cells through increasing lipid peroxidation and decreasing ALDH3A1 expression [56]. Additionally, TZD-mediated PPARγ activation inhibited ALDH1A3 expression to exert anti-proliferative functions in H1993 cells [57].

### 6.2. Promotion of Cell Apoptosis

Apoptosis is a homeostatic mechanism to maintain cell populations in normal tissues, whereas tumor cells engage various mechanisms to evade apoptosis for unrestricted proliferation [58,59]. Classical pathways of apoptosis include the intrinsic mitochondrial pathway and the extrinsic pathway that induces via the activation of death receptors on the cell surface, both of which result in the activation of cysteine aspartyl-specific proteases (also known as Caspases) to cleave several proteins leading to cell death [60,61]. PPARγ promotes apoptosis in lung cancer through dysregulating critical factors in these pathways (Figure 2). Specifically, PPARγ activation could increase the expression of pro-apoptotic factors Bax and Bad, decrease the expression of anti-apoptotic factors Bcl-2 and Bcl-XL, enhance caspase3 and caspase9 activity, and trigger mitochondrial cytochrome c release [62,63,64,65,66,67]. Furthermore, PPARγ activation enhanced TRAIL-induced apoptosis in human lung adenocarcinoma cells via autophagy flux [68]. Notably, PPARγ is also able to regulate the upstream signaling pathways to provoke apoptosis. For instance, KR-62980 or rosiglitazone-mediated PPARγ activation promoted the generation of reactive oxygen species (ROS) via proline oxidase (POX) induction, leading to apoptotic cell death in NSCLC [69]. Troglitazone-mediated PPARγ activation induced the phosphorylation of extracellular signal-regulated protein kinases 1 and 2 (ERK1/2) and subsequently caused apoptosis in NCI-H23 lung cancer cells via a mitochondrial pathway [70]. Another study showed that troglitazone-mediated PPARγ activation stimulated the expression of DNA-damage inducible gene 153 (GADD153) to trigger growth arrest and endoplasmic reticulum stress-induced apoptosis of NSCLC cells [71]. PPARγ agonist efatutazone could induce the cell cycle arrest and apoptosis of EGFR-TKI-resistant lung adenocarcinoma cells via PPARγ/phosphatase and tensin homolog (PTEN)/Akt pathway [72].

### 6.3. Induction of Cell Cycle Arrest

Cell division is tightly regulated by multiple conserved cell cycle control mechanisms, such as cyclins and cyclin-dependent kinases (CDKs), G1-S transcriptional regulation, DNA damage checkpoint, DNA replication stress checkpoint, and spindle assembly checkpoint [73]. The eukaryotic cell cycle can be divided into G0, G1, S, G2, and M phases. DNA replicates during the S phase and cell separates during the M phase. Several lines of evidence have shown that PPARγ is involved in cell cycle processes to induce the growth arrest of lung cancer cells (Figure 2). Troglitazone-mediated PPARγ activation could induce cell cycle arrest in the G0/G1 phase by downregulation of cyclins D and E [74]. The PPARγ ligands PGJ2, ciglitazone, troglitazone, and GW1929 suppressed human lung carcinoma cell growth by stimulating cyclin-dependent kinase inhibitor p21 expression and reducing cyclin D1 expression [75].

### 6.4. Inhibition of Tumor Metastasis

Metastasis is the cause of 90% of cancer-related deaths [76]. In the process of metastasis, normal cells transform into carcinogenic cells that proliferate uncontrollably, evade the immune system, resist programmed cell death, stimulate angiogenesis, acquire invasive potential, survive in the bloodstream, and establish cancerous growth in distant organs [77]. Epithelial–mesenchymal transition (EMT), a process through which epithelial cells lose apical–basal polarity, and cell–cell junctions, as well as attain a mesenchymal phenotype with invasive and migratory capabilities, is a critical event in the initiation of metastasis [77,78]. The mechanism of PPARγ in lung cancer-related EMT is not yet fully understood. Multiple studies have shown that PPARγ can regulate the expression of EMT-related molecules to exert an inhibitory effect on metastasis (Figure 2). Specifically, PPARγ activation increased expression of epithelial marker E-cadherin, decreased expression of mesenchymal marker N-cadherin, Snail, and fibronectin, as well as down-regulated expression of matrix metalloproteinase 9 (MMP-9) and heparanase (HPA) [79,80,81,82,83]. Moreover, PPARγ activation could also suppress the expression of invasion-related proteins, such as intercellular adhesion molecule-1 (ICAM-1) and C-X-C chemokine receptor type 4 (CXCR4), which function as facilitators in EMT [80,81,83].

Angiogenesis is the major route by which cancer cells spread from the primary tumor to other sites [77]. Tumor angiogenesis is regulated by both pro- and anti-angiogenic factors, and an imbalance between the two can lead to malformation of the vasculature with excessive branching, hyperpermeability, and leakage [77,84]. PPARγ has been reported to be involved in regulating angiogenic factors (Figure 2). For instance, PPARγ activation could inhibit angiogenesis by blocking the production of ELR + CXC chemokines in NSCLC [85]. Moreover, vascular endothelial growth factor (VEGF) was drastically downregulated through the PPARγ/NF-κB signaling pathway in human lung carcinoma 95D cells [64].

### 6.5. Influence on Tumor Immunity

The role of PPARγ activation in lung cancer immunity remains a controversial issue, which may correlate with the complexity of the immune microenvironment. Notably, the tumor microenvironment encompasses not only malignant cells, but also stromal cells, vascular endothelial cells, as well as various types of immune cells including tumor-associated macrophages and myeloid-derived suppressor cells (MDSCs) [86]. Interestingly, PPARγ seems to have opposing effects on cancer progression among different cells, with anti-oncogenic effects on cancer cells but pro-oncogenic effects on cancer-associated immune cells [87,88]. Gou et al. found that PPARγ inhibits the tumor immune escape by inducing PD-L1 autophagic degradation in NSCLC cells [89]. Interestingly, this process was independent of the transcriptional activity of PPARγ, but rather formed autophagy receptors through the binding of PPARγ to microtubule-associated protein 1A/1B-light chain 3 (LC3), leading to degradation of PD-L1 in lysosomes. Nevertheless, Li et al. suggested that PPARγ activation in myeloid cells promoted lung cancer progression and metastasis [87].

## 7. Therapeutic Exploration of PPARγ Agonists

Given the multiple functions of PPARγ activation in lung cancer, PPARγ agonists included thiazolidinediones and non-thiazolidinediones have been used as therapeutic agents to tackle lung cancer in preclinical and clinical studies.

### 7.1. Thiazolidinediones

Thiazolidinediones (TZDs), also known as glitazones, were first reported as insulin sensitizers in the early 1980s and were found to be ligands for PPARγ until the 1990s [90,91]. Ciglitazone is the prototype of all TZDs, but has never been approved for clinical application due to its weak therapeutic effect on diabetics. Troglitazone was the first TZD introduced in 1997, but was quickly removed from the market in 2000 because of its serious hepatotoxicity [92]. Both rosiglitazone and pioglitazone were second-generation TZDs and were released in 1999. However, rosiglitazone was temporarily withdrawn due to a connection to adverse cardiovascular effects, while pioglitazone was restricted in light of a possible increased risk of bladder cancer [93,94]. Efatutazone is a novel third-generation TZD with highly selective and is currently undergoing clinical evaluation. Although TZDs were developed as an anti-diabetic drug and known to cause side effects, the potent PPARγ activating effects of TZDs have driven extensive exploration of their potential as anti-cancer therapies for lung cancer. TZDs exert anti-lung cancer functions in PPARγ-dependent and PPARγ-independent manners, of which PPARγ-dependent effects have been summarized in the above section.

Several studies have shown that PPARγ antagonist or siRNA-mediated silencing of PPARγ expression failed to abrogate certain anti-tumor effects of TZDs. Han et al. provided evidence that rosiglitazone inhibited NSCLC cell proliferation via down-regulation of the Akt/mTOR/p70S6K signaling pathway [95]. Moreover, Sun et al. found that nicotine-induced NSCLC cell proliferation was partly mediated through alpha4 nAChR, which could be blocked by rosiglitazone through activating the ERK/p38 MAPK/p53 signaling pathway in a PPARγ-independent manner [96]. Zou et al. reported that the PPARγ ligands troglitazone, cigolitazone, and GW1929 exerted PPARγ-independent effects to upregulate expression of death receptor 5 and downregulate c-FLIP levels, thereby enhancing TRAIL-induced apoptosis [97]. Similarly, ciglitazone could inhibit NSCLC cell proliferation in a PPARγ-independent mechanism. Hann et al. found that ciglitazone suppressed the expression of phosphoinositide-dependent protein kinase 1, which was not blocked by GW9662, leading to the inhibition of the growth of NSCLC cells [98].

### 7.2. Other Agonists

Non-TZD PPARγ agonists have also been explored the potential of the treatment of lung cancer. 13-S-hydroxyoctadecadienoic acid (13(S)-HODE) and 15(S)-hydroxyeicosatetraenoic acid (15(S)-HETE), as endogenous ligands for PPARγ, were significantly reduced in NNK-induced lung cancer and could inhibit NSCLC when exogenously supplemented [99,100]. KR-62980, a selective PPARγ agonist, induced apoptotic cell death in NSCLC mainly through ROS formation via POX induction [39,69]. Telmisartan, a partial agonist with a benzimidazole scaffold, inhibited the expression of ICAM-1 and MMP-9 to exhibit an anti-proliferative effect in A549 cells [80]. Bavachinin, a natural bioactive flavanone from *Psoralea corylifolia*, induced the death of A549 cells through mediating ROS generation [62,101]. All of CB11, CB13, and PPZ023, designed by Kim et al., were suggested to overcome the radioresistance of lung cancer [66,67,102].

### 7.3. Potential for the Combination of PPARγ Agonists and Other Therapies

Therapeutic resistance remains a major obstacle to achieving cures in patients with cancer [103]. The discovery of epidermal growth factor receptor (EGFR) gene alterations in lung cancer has fueled the development of targeted therapy using tyrosine kinase inhibitors (TKIs) [104]. EGFR-TKIs act as a first-line treatment for patients with EGFR mutation; however, most patients fail to gain sustainable benefit due to developing resistance [105]. In preclinical studies, PPARγ has been demonstrated to exert synergistic therapeutic potential with EGFR-TKIs. Lee et al. suggested that rosiglitazone potentiated the antiproliferative effects of gefitinib by increased PTEN expression [106]. Serizawa et al. reported that efatutazone inhibited cell motility by antagonizing the TGF-β/Smad2 pathway and effectively prevented metastasis in NSCLC patients with acquired resistance to EGFR-TKI [107]. Another study completed by Ni et al. found that efatutazone and gefitinib synergistically inhibited the proliferation of EGFR-TKI-resistant lung adenocarcinoma cells via the PPARγ/PTEN/Akt pathway [72]. To et al. suggested that PPARγ agonists enhanced the anti-cancer effects of gefitinib through activating the PTEN/PI3K/Akt signaling pathway [108]. Furthermore, PPARγ can also increase the efficacy of conventional chemotherapy and radiotherapy. Specifically, troglitazone could synergize with cisplatin or paclitaxel to inhibit NSCLC both in vitro and in vivo in a sequence-specific manner, while rosiglitazone combination with carboplatin reduced the growth of KRAS- or EGFR-mutated lung cancers [109,110]. In addition, PPARγ agonists, including ciglitazone, PPZ023, and CB13, could function as a radiosensitizer in radioresistance-related lung cancer through inducing ROS generation or ER stress [66,67,111].

### 7.4. Clinical Trials

A retrospective analysis of 87,678 male diabetics demonstrated that TZD users showed a 33% reduction in the risk of lung cancer compared with nonusers [112]. Subsequently, several clinical trials were launched to test the efficacy of PPARγ agonists in the treatment of lung cancer. Wigle et al. suggested a potential preventive effect for pioglitazone in early stage NSCLC through a clinical trial in patients with stage IA-IIIA NSCLC [113]. Nicotine exposure remains a major risk factor for lung cancer [2]. Keith et al. conducted a phase II trial to investigate pioglitazone as a chemoprevention for lung cancer in high-risk smokers and found that pioglitazone could slightly improve endobronchial dysplasia [114]. Jones et al. verified that pioglitazone could reduce nicotine craving in heavy smokers [115]. This interesting study indicated that pioglitazone might exert its anti-lung cancer effects via multiple mechanisms, rather than only targeting malignant cells. Indeed, pioglitazone-mediated PPARγ activation has been reported to attenuate the expression of physical and emotional nicotine withdrawal symptoms through mechanisms involving amygdala and hippocampal neurotransmission [116]. Furthermore, a preclinical study identified that glitazone could inhibit nicotine-induced inflammation via downregulation of the Toll-like receptor 4 signaling pathway in alveolar macrophages [117].

## 8. Conclusions and Prospects

PPARγ agonists have shown beneficial effects in anti-lung cancer, including disruption of tumor metabolic homeostasis, promotion of cell apoptosis, induction of cell cycle arrest, as well as inhibition of cell invasion and angiogenesis. However, the role of PPARγ in tumor immune microenvironment is still controversial. Specifically, PPARγ induced PD-L1 degradation in malignant cells, which contributes to enhancing the sensitivity to immunotherapy, whereas PPARγ activation in myeloid cells promotes lung cancer progression and metastasis. Therefore, it is essential to evaluate the systemic effects of PPARγ agonists in tumor sites. Even so, PPARγ remains a viable target for the treatment and prevention of lung cancer due to the effectiveness of PPARγ agonists as monotherapy and in combination with traditional radiotherapy or chemotherapy in preclinical studies. Notably, although TZDs have some side effects, they are the most potent PPARγ agonists and have been conducted to explore anti-lung cancer effects in clinical settings. While most partial agonists with fewer side effects were initially developed for the treatment of metabolic diseases, their gene expression signatures could also be optimized to provide more anti-tumor benefits in the future. Herein, we summarized the molecular mechanisms of action of agonists and the complex signaling networks resulting from PPARγ activation, which may contribute to the design of PPARγ agonists characterized by more efficient, safer, and potent anti-tumor effects in the future.

## Figures and Tables

**Figure 1 biomolecules-14-00190-f001:**
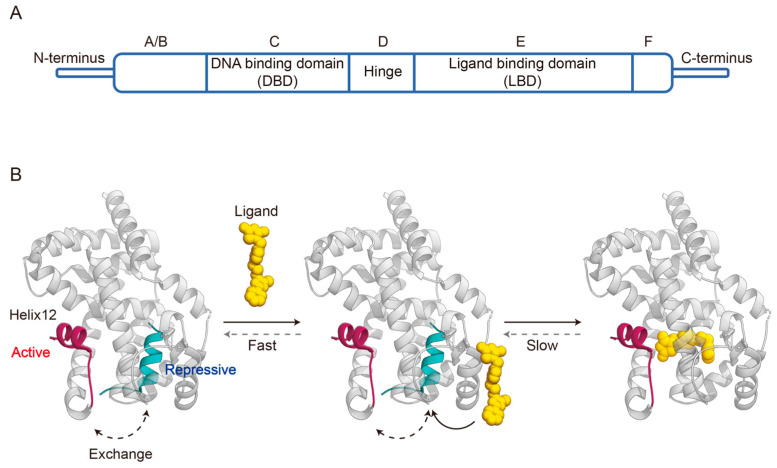
Domain organization of PPARγ (**A**) and schematic of agonist binding with PPARγ LBD (PDB code: 6ONJ and 6DGL) (**B**).

**Figure 2 biomolecules-14-00190-f002:**
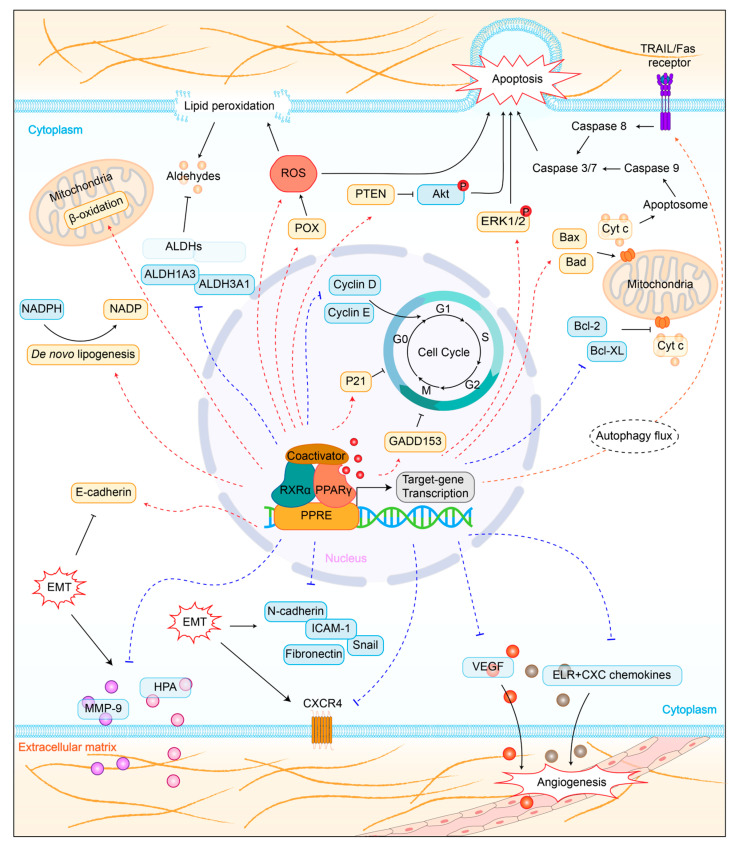
Role of PPARγ activation in lung cancer cells. PPARγ, peroxisome proliferator-activated receptor gamma; RXRα, retinoic X receptor α; PPRE, PPAR response elements; GADD153, DNA-damage inducible gene 153; NADPH, nicotinamide adenine dinucleotide phosphate; ALDH, aldehyde dehydrogenases; ROS, reactive oxygen species; POX, proline oxidase; PTEN, phosphatase and tensin homolog; ERK1/2, extracellular signal-regulated protein kinases 1 and 2; cyt-c, mitochondrial cytochrome c; EMT, epithelial-mesenchymal transition; MMP-9, matrix metalloproteinase 9; HPA, heparanase; ICAM-1, intercellular adhesion molecule-1; VEGF, vascular endothelial growth factor.

**Table 1 biomolecules-14-00190-t001:** Chemical structure and transactivation of common PPARγ ligands.

Class	Compound	Structure	Transactivation (EC_50_)	Refs.
Natural ligand	15-deoxy-D12,14-prostaglandin J2	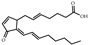	0.42 μM	[36]
Full agonist	Ciglitazone	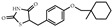	3 μM	[37]
Troglitazone	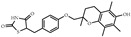	0.55 μM	[37]
Pioglitazone	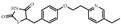	0.58 μM	[37]
Rosiglitazone	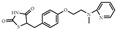	43 nM	[37]
Efatutazone	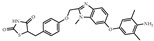	0.038 nM	[38]
Partial agonist	KR-62980	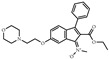	15 nM	[39]
Telmisartan	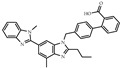	4.3 μM	[40]
Inverse agonist	T0070907	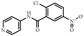	1.0 nM	[41]
Antagonist	GW9662	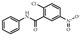	3.3 nM	[41]

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
