# Peer review of "PPARγ Modulators in Lung Cancer: Molecular Mechanisms, Clinical Prospects, and Challenges"

_biomolecules, 2024, doi:10.3390/biom14020190_

Round 1
Reviewer 1 Report
Comments and Suggestions for Authors
The review is well written and already of interest. I however have some suggestions to make it more complete.
1) in the nuclear receptor field we always use the official nomenclature the first time we introduce the full name of a nuclear receptor, hence p1 L44 put in bracket that PPARgamma is known as NR1C3
2) PPAR gamma ligands: the authors cited PGJ2 as a PPARgamma ligands but the consensus in the field is that PGJ2 does have more PPAR gamma independent effects than PPAR gamma mediated effects and seems to be produced at low levels, which are not physiologically relevant (PUBMED ID 12975479). Please cite this work and explain briefly the problem for the sake of science.
3) PPAR gamma ligands: the authors did not cite some natural ligands such as 15(S)-HETE and 13(S)-HODE which are known to be reduced in lung cancer and as such reduced PPAR gamma activity in the cancer context (PUBMED ID 20388757)
4) PPAR gamma ligands and lung cancer: the authors could speculate on the effects of other ligands like glitazars that are co activating PPAR alpha and gamma? It is intriguing for instance that PPARα and PPARγ activation associated with pleural mesothelioma invasion but that therapeutic inhibition was ineffective (PUBMED ID 34984327). They should also comment on the role of RXR ligands knowing that RXR is activated by rexinoids coming from vitamin A pathway and that vitamin A intake must be low in smokers to not produce oxidative stress. Would it make sense to co activate PPARgamma and RXR to try to maximize effects in clinical trials? Several studies showed that vitamin A and carotenoids intake in smokers correlate with lung cancer development. Finally what would the authors expect from partial agonist like for instance the old LG100754 which is an interesting drug in terms of PPAR gamma activation? It was a bit disappointing to see SPPARM cited but not discussed.
5) Given the role of cigarette smoke in lung cancer I was expecting a bit more of references and discussion when the authors described clinical trials and the effects of glitazones on lung cancer development. Indeed, glitazones in diabetics seem to protect partially from lung cancer development as written p10 L339, I would like to add that multiple mechanisms are likely at work with pioglitazone reducing craving for nicotine (PUBMED ID 29020601), probably via amygdala and hippocampus neurotransmission (PUBMED ID 31685649), but also that glitazone by lowering TLR4 in alveolar macrophages counteract local inflammation induced by smoking (PUBMED ID 24612634) and as such the effects of PPARg agonists are not only targeting one single cell compartment.
6) One key target gene of PPAR gamma is CD36 a fatty acid transporter, interestingly CD36 expression was found in SCLC as shown recently in Cell (PUBMED ID 38181741) could the author speculate on the implication of CD36 and its activation
Reviewer 2 Report
Comments and Suggestions for Authors
An interesting review article containing a fairly large amount of relevant scientific material on the problem of "Role of PPARy pathways in lung cancer cells". There are the following suggestions and questions about this article:
1. When considering the characteristics of the PPARG gene, which controls the formation of various PPARγ1 (encoded by PPARG1, PPARG3, PPARG4 mRNAs) and PPARγ2 (encoded by PPARG2 mRNA) isoforms it is recommended to provide data (may be in a separate table) on functionally significant polymorphisms of this gene, which can affect the level of expression/splicing of this gene and, accordingly, will affect the final phenotypic effects of PPARy in lung cancer.
2. It is recommended to provide data (may be in a separate table) on the relationship of functionally significant polymorphisms for PPARy with lung cancer (or other oncological diseases). May be there are data from genome-wide studies that show the association of PPAGy polymorphisms with lung cancer or other oncological diseases?
Round 2
Reviewer 2 Report
Comments and Suggestions for Authors
All necessary explanations are given by the authors. The article is recommended for publication.